# Early Identification and Treatment of Trochlear Knee Dysplasia

**DOI:** 10.3390/jpm13050796

**Published:** 2023-05-06

**Authors:** Joaquin Moya-Angeler, Cristina Jiménez-Soto, Domingo Maestre-Cano, Carlos de la Torre-Conde, Regina M. Sánchez-Jimenez, Cristina Serrano-García, Miguel Alcaraz-Saura, Juan Pedro García-Paños, César Salcedo-Cánovas, Francisco Forriol, Vicente J. León-Muñoz

**Affiliations:** 1Department of Orthopedic Surgery, Hospital Universitario Reina Sofia, 30003 Murcia, Spain; jmoyaangeler@gmail.com; 2Instituto de Cirugía Avanzada de Rodilla, 30005 Murcia, Spain; 3School of Medicine, Murcia University, 30120 Murcia, Spain; cristinajmsoto@gmail.com; 4Department of Orthopedic Surgery, Hospital Clínico Universitario Virgen de la Arrixaca, 30120 Murcia, Spain; domingomsc@gmail.com (D.M.-C.);; 5Department of Orthopedic Surgery, Hospital de la Ribera, 46600 Alzira, Spain; carlosdelatorre94@gmail.com; 6Department of Pediatric Radiology, Hospital Clínico Universitario Virgen de la Arrixaca, 30120 Murcia, Spain; refgi@hotmail.com (R.M.S.-J.); serrano5977@gmail.com (C.S.-G.); 7Department of Pediatrics, Hospital Clínico Universitario Virgen de la Arrixaca, 30120 Murcia, Spain; miguelalcarazs@gmail.com; 8School of Medicine, University CEU-San Pablo, 28668 Alcorcón, Spain; fforriol@mac.com

**Keywords:** trochlear dysplasia, patellar instability, Pavlik harness, breech presentation

## Abstract

A shallow sulcus characterizes trochlear dysplasia (TD) of the femoral trochlea, which can lead to chronic pain or instability of the patellofemoral joint. Breech presentation at birth has been identified as a risk factor for developing this condition, which an ultrasound can identify early. Early treatment could be considered at this stage, given the potential for remodelling in these skeletally immature patients. Newborns with breech presentation at birth who meet the inclusion criteria will be enrolled and randomised in equal proportions between treatment with the Pavlik harness and observation. The primary objective is to determine the difference in the means of the sulcus angle between the two treatment arms at two months. Ours is the first study protocol to evaluate an early non-invasive treatment for TD in the newborn with breech presentation at birth using a Pavlik harness. We hypothesised that trochlear dysplasia could be reverted when identified and treated early in life with a simple harness, as it is done with developmental dysplasia of the hip.

## 1. Introduction

The femoral trochlea consists of a groove or sulcus through which the patella slides during motion. This groove is the main stabilizer of the patellofemoral joint above 30° of knee flexion. Below 30°, the main stabilizer is the medial patellofemoral ligament (MPFL). The abnormal morphology of the trochlear groove characterizes trochlear dysplasia (TD). This groove has a physiologic concavity with a normal angle of 135 ± 10° [1,2]. If the morphology of the trochlear groove becomes flatter or convex, as is characteristic of TD, the patella is not properly stabilized during knee flexion.

TD is one of the most important contributors to patellar instability [3], presenting in 96% of patients with a history of patellar dislocation [4]. TD begins manifesting in adolescence and early adulthood. This condition has also been associated with anterior knee pain with or without maltracking [5] and instability [1]. These abnormal biomechanics might, in addition, lead to degenerative cartilage changes and patellofemoral osteoarthritis [6,7,8]. Considering this entity’s repercussions, it seems reasonable to apply a corrective treatment in more severe cases to treat recurrent dislocation episodes and prevent future patellofemoral degenerative complications. The mainstay treatment for instability with trochlear dysplasia type B and D of Dejour classification [9] is trochleoplasty, which can be associated with other additional gestures such as MPFL reconstruction and tibial tubercle transfers, among others. In childhood, trochleoplasty is not free of controversy. If necessary, a deepening trochleoplasty can be indicated in patients with high-grade trochlear dysplasia, with an expected growth of fewer than two years. This procedure has ultimately been shown to be able to reshape the trochlea in the skeletally immature patient [10,11,12,13]. However, this potential is limited in children older than ten years, and the procedure is technically demanding and not exempt from several potential complications [14]. Therefore, no current treatment allows trochlear remodelling to occur during this period, and this condition’s late manifestation makes an early diagnosis during childhood unlikely.

The aetiology of TD remains unclear. The articular surface of the sulcus is developed at birth. The osseous sulcus angle is inversely related to age, being flattest in the youngest children and deepening steadily through growth. By contrast, the cartilaginous sulcus angle stays virtually constant from birth to adolescence [15,16].

Some authors argue for the genetic origin of trochlear dysplasia [17,18], as the morphology of the distal femur seems to be the same in the foetus and the adult. Moreover, in congenital patellar dislocation, the sulcus is altered [19,20,21]. Another hypothesis contemplates that biomechanical factors could influence the final shape of the trochlear groove. Recurrent episodes of patellar dislocation in skeletally immature patients have been shown to lead to the development of TD in those without previous congenital abnormalities [8,10,11,22,23]. Additionally, experimental models on rabbits [10,11,23,24] have shown that an inadequate patellar position predisposes the distal femoral groove to become more flattened. This abnormal patellar position could modify the mechanical stress distribution, altering cartilage growth in immature skeletons or affecting its metabolism when mature, resulting in cartilage damage and early-onset osteoarthritis [6].

According to Frost’s theoretical cartilage growth force response curve [25], mechanical load partially regulates chondral growth. Decreased compression limits articular chondral growth because cartilage growth directly relates to compression until excessive forces beyond physiologic loads are reached that first retard and then stop growing. Therefore, when a foetus is in the breech presentation, an abnormal interaction between the patella and the trochlea could result in an intrauterine remodelling of a previously normal foetal knee.

The breech presentation is associated with TD [26,27,28], with a higher incidence in the frank breech position [28] where both legs are in extension. TD has also been related to patella alta [24,29]. Footling and frank breech positions could act similarly, resulting in a higher, more proximal patellar position, preventing it from engaging with the femoral trochlea. The decreased contact pressures result in diminished compression between the surfaces, and chondral tissue growth decreases, resulting in an abnormal trochlear groove depth.

During growth, cartilage is replaced with bone. Therefore, chondral remodelling could correct a deformity in skeletally immature patients [25]. To correct this condition, normal load distribution would need to be re-established. The patella enters the trochlear groove at the beginning of knee flexion, at approximately 30°, and the contact surface increases until it reaches its maximum at 90° [10]. Therefore, if the knee is maintained in this position for a period that allows remodelling, it would allow the maximum peak of physiologic compression to be exerted, stimulating cartilage growth and remodelling. For this purpose, the Pavlik harness could be used.

The Pavlik harness is widely used to treat developmental dysplasia of the hip (DDH) with a time-dependent correction success rate. However, the maximum period for treatment with the Pavlik harness has yet to be established [30]. Usually, the recommended period is a maximum of eight weeks for non-displaced hips [31]. Even though the hip and knee are different articulations, Frost’s cartilage growth principle [25] could similarly be applied. Since this is the first study to employ the Pavlik harness in TD, the ideal treatment period for this condition is unknown. Therefore, we will apply the eight weeks timeline for the treatment of TD with the Pavlik harness.

If TD could be early reverted with a simple harness in the newborn, as it is nowadays done with DDH, this could potentially normalize the patellofemoral joint biomechanics, increasing stability, enabling proper function, and preventing pain and early-onset degenerative arthropathy. Observation is the usual treatment for the first episode of dislocation in the pediatric patient. Patellofemoral instability often responds poorly to conservative treatment, and dislocation repeatedly recurs in a significant number of patients with risk factors for dislocation [32]. TD can be aggravated in patients with an immature skeleton experiencing recurrent episodes of dislocation, ultimately leading to an increase in instability, risk of cartilage deterioration, pain, and reduced health-related quality of life and financial costs [33]. MPFL reconstruction is usually the surgical technique of choice in paediatric patients. However, it does not avoid the risk of redislocation [34], especially when TD and other risk factors are present.

We therefore propose an early non-surgical and non-invasive treatment in newborns with an abnormal ultrasound femoral sulcus angle, applying a Pavlik harness for two months. We hypothesize that acquired factors in the development of the femoral trochlea are essential and that the femoral dysplastic trochlea can remodel early in life when appropriately identified and treated. We aim to provide a therapeutic option at the early stages of development to revert TD.

## 2. Materials and Methods

### 2.1. Methods: Participants, Interventions, Outcomes

The study is a registered clinical trial with ClincalTrials.gov ID: NCT05081453.

#### 2.1.1. Trial Design

The study is designed as a prospective, randomized, controlled, triple-blinded, superiority clinical trial with two parallel groups and a primary endpoint of trochlear dysplasia screening and treatment over two months. Randomization will be performed as block randomization with a 1:1 allocation.

#### 2.1.2. Study Setting

The present investigation will be conducted in the Orthopedic Surgery Department of Virgen de la Arrixaca University Hospital in Murcia (Spain), a level III Academic Hospital. For data and clinical outcomes, the Pediatric Radiology and Pediatric Orthopedic Surgery Departments will follow up with patients. The principal investigator and research assistant will perform data management and statistical coordination.

#### 2.1.3. Eligibility Criteria

Inclusion criteria:Healthy neonates;Breech presentation;Sulcus angle ≥ 159° in ultrasound;

Exclusion criteria:Older than three months of age;Cerebral palsy;Developmental dysplasia of the hip: pathological alpha angle according to Graf classification (except type IIa) in ultrasound, or clinical criteria: positive Barlow’s or Ortolani’s test, hip clicks, tightness of adductors, asymmetry of the skin folds;Developmental alterations;Chromosomopathies;Other pathologies that could be adversely affected by the introduction of the Pavlik harness.

#### 2.1.4. Interventions

Eligible patients will be randomized in equal proportions between treatment with the Pavlik harness and observation, as shown in Figure 1.

The pediatric orthopedic department will provide the harness in its commercially available form (ORLIMAN Pavlik Harness CE).

The harness will be placed with a hip flexion of 60° and a knee flexion of 90°, as shown in Figure 2. The Pavlik harness’s first placement will occur after clinical exploration by a pediatric orthopedic surgeon and an ultrasound exam. The pediatric radiologist performing the ultrasounds, the data collectors, and the data analysts will be blinded to the identity of the intervention group patients.

#### 2.1.5. Modifications

The patient will be monitored for concomitant pathologies that have arisen during treatment with the Pavlik harness to assess whether they will cause discomfort or difficulty for the patient and their family. In case of incompatibility, treatment with the Pavlik harness will be withdrawn.

#### 2.1.6. Adherence

A face-to-face evaluation of the subjects in the treatment group will be conducted after four weeks to assess the correct use of the harness.

It is beneficial to check daily the flexion creases of the knees, groin, and neck to identify possible chafing of the skin, in which case soft padding is applied to cover the straps.

During the first visit, the patient’s parents will learn recommendations for daily situations such as nappy changing, dressing, resting, breastfeeding, playing, and bathing at home. They will also receive a complete written guide. On the second visit, the researchers will interview the parents to learn about daily life adaptations such as breastfeeding, play, nappy changing, and dressing.

#### 2.1.7. Concomitant Care

Relevant concomitant care and interventions permitted or prohibited during the trial are detailed in the written guide provided to the treatment group patient’s caregivers. In this guide, they will also have a phone number to reach out in case questions need further clarification.

#### 2.1.8. Outcomes

Outcomes measures will be collected two months after the first visit.

Primary outcome measures:The difference in the means of the sulcus angle between the two treatment arms at two months.

Treatment success is defined as a sulcus angle < 159° visualized by ultrasound. A sulcus angle ≥ 159° is the value set to diagnose TD in our study population, and values < 159° are considered non-dysplastic.

For newborns, radiographs and CT are not optimal imaging modalities. Besides X-ray radiation, the patellofemoral joint is not ossified, so it is not visible [36]. MRI is not an optimal screening tool because of the need for sedation, limited availability, and costs. Ultrasound allows for the visualization of cartilage and bone and is a faster, cheaper, and more available technique, so it could be used as a screening tool for detecting TD [37]. An ultrasound sulcus angle > 159° is considered dysplastic in newborns [28].

In our study, the ultrasound will be performed by a single observer in all cases to eliminate any possible inter-observer variability.

Secondary outcome measures:The difference in the means of the sulcus angle at baseline and after two months for each group;The difference will be measured by comparing the mean sulcus angle before and after the two-month treatment for the intervention group and the baseline and after two months for the control group.

The difference between the two treatment arms will be measured by comparing the proportion of patients in each treatment group with dysplastic (≥159°) and non-dysplastic (<159°) sulcus angles.

#### 2.1.9. Participant Timeline

During the first four to six weeks of life, a bilateral knee ultrasound will be performed on neonates with a breech presentation to determine the sulcus angle. Then, they will be derived into the Pediatric Orthopedics and Traumatology Department, as stated in Figure 2.

Intervention group: The estimated duration of the Pavlik harness procedure is two months, with four-week revisions by a pediatric orthopedic surgeon. After two months, a pediatric radiologist will perform a second ultrasound to evaluate the sulcus angle, and the Pavlik harness will be removed. In both groups, a face-to-face consultation in pediatric orthopedic surgery will be performed at 6–9 months to assess the clinical progression, as shown in Figure 3.

Control group: the control group will be evaluated two months after the baseline ultrasound, with a second one performed by a pediatric radiologist, as shown in Figure 4.

#### 2.1.10. Sample Size

The study population are the newborns at TD of Virgen de la Arrixaca University Hospital. The total number of births in this institution in 2016 was 14,167. However, the frequency of TD in newborns is unknown.

Breech presentation is a risk factor for TD, and the frequency of breech presentation is between 3–5% [38]. We consider 5% to be the estimated frequency of this presentation at birth, which implies that approximately 708 newborns each year have a breech presentation in our population. TD (a pathologic sulcus angle) is identified in 10% of the breech presentation population [28]. Accordingly, we consider the number of newborns in our population with TD to be 71 per year.

For the ultrasound screening test, we need 116 ultrasounds to have a 95% confidence that the real value is within ±5% of the measured one. This estimation was calculated according to the following formula:$$n=\frac{\;\frac{Z_{\alpha }^{2}\cdot p\cdot \left(1-p\right)}{e^{2}}}{1+\left(\frac{Z_{\alpha }^{2}\cdot p\cdot \left(1-p\right)}{e^{2}\cdot N}\right)}$$
where *Z_α_* is the critical value of the normal distribution (for a confidence level of 95%, *α* is 0.05, and the critical value is 1.96), *e* is the margin of error (*e* = 5%), *p* is the sample proportion (*p* = 10%), and *N* is the population size (*N* = 708). Note that a finite population correction has been applied to the sample size formula.

For the estimation of the treatment groups sample size, the following formula is applied: $$n=\frac{2{\left(a+b\right)}^{2}\;\sigma^{2}}{{\left(\mu_{1}-\mu_{2}\right)}^{2}}$$
where *n* is the sample size in each group, a is the conventional multiplier for alpha, b is the conventional multiplier for power, σ is the sample variance, and *μ*_1_ − *μ*_2_ is the difference we aim to detect. The sample variance was calculated from a pilot study, from which we obtained a value of *σ* = 5.78.

To consider a difference of 10° in sulcus angles between the treatment and the control group as clinically significant (*μ*_1_ − *μ*_2_) and be detected with 80% power and a significance level alpha of 0.05, *n* = 5.25. That means six subjects per group is the minimum sample size.

#### 2.1.11. Recruitment

The sample of choice will be neonates with the breech presentation during late pregnancy, after screening for inclusion/exclusion criteria and obtaining informed consent.

### 2.2. Methods: Assignment of Interventions

#### 2.2.1. Allocation

Once the sulcus angle measurements have been studied by an ultrasound, a random block sampling will be performed with the “Study Randomizer” software (https://www.studyrandomizer.com/ accessed on 14 December 2022). The pediatric orthopedic surgeon will index the patient history number that identifies each subject in this software. The software then generates a hidden sequence, which assigns a group to each subject. The analyst will also be blinded by the hidden assignment sequence used by the software.

#### 2.2.2. Blinding

Patients’ caregivers cannot be blinded to the treatment with the Pavlik harness. However, the patient will attend the ultrasound tests and the pediatric orthopedic surgery check-ups without the harness. The clinical analyst will be blinded by the software that keeps the assignment sequence associated with each subject hidden.

### 2.3. Methods: Data Collection, Management, and Analysis

#### 2.3.1. Data Collection Methods

Ultrasound examination will be performed with the patient in a supine position and the knee placed at 45–90° of flexion. The positioning of the transducer must be in a transversal plane perpendicular to the axis of the femoral diaphysis. This way, the femoral condyles, the trochlea, and the ossification centre are visible.

Likewise, the following variables will be documented:Age;Sex;Birth weight;Birth presentation (footling, frank or complete breech);Presentation at the last ultrasound performed during pregnancy;Coxofemoral joint status according to clinical (Graf classification [39]: I = normal, II = unstable, III and IV = off-centre) and sonographic parameters (α angle).

Based on the review by the pediatric orthopedic surgeon, the following variables will be controlled:Verification of the suitable position of the harness during the check-ups performed every four weeks until two months;Treatment difficulties and adverse reactions: blistering, inadaptation, crying, femoral nerve palsy, avascular necrosis of the femoral head;The discrepancy of inferior limb length;Skin folds asymmetry;Knee articular balancing: complete and symmetrical, uncomplete of one knee, uncomplete of both knees;Articular click;Patellar displacement: tolerated, apprehension in the form of crying or discomfort.

#### 2.3.2. Data Management

All relevant data of the study subjects will be transcribed with a blue pen to the Data Recollection Notebook (DRN). The DRN must be filled entirely and in legible handwriting to later perform the statistical analyses. Once the study is finished, all data collected will be evaluated for the final report writing.

All the study documentation will remain stored in the researcher’s archive of the participant centre, in the custody of the principal investigator (PI) until the finalization of the study. Once finished, the documents will be indexed and stored in the centre’s general archive, according to the recommendations of good clinical practice.

The PI will take care of the conservation of the identification codes for at least fifteen years after the study is finished or interrupted.

The promoter or the data owner will preserve all the documentation relative to the study for the time required by the law.

In any case, the confidentiality of the data and documents in the file will be ensured.

Data from all DRNs will be entered into a database created for this purpose and provided with security margins and internal consistency rules. This database shall be equipped with a double-entry system and filters to prevent and detect any inconsistencies or errors in the database. The information will be validated by internal consistency checks, looking for missing values. Data will be checked and corrected until the database is fully validated.

Once the database is cleaned, the variables will recode by generating new variables (regroupings, summations, etc.).

#### 2.3.3. Statistical Methods

The primary efficacy endpoint is the comparison of the treatment with the Pavlik harness for two months, measured by analyzing the initial ultrasound sulcus angle data and those obtained after the treatment period.

The Shapiro–Wilk test will be performed to assess the normality of data.

To determine if the Pavlik harness is more effective in reducing knee SA compared to no treatment, the results of the intervention group will be compared to the control group with the Student’s t-test for independent data. The intervention will be effective if it achieves statistical differences (with a 95% confidence) in the knee SA between the treatment and control groups before and after treatment.

To assess if the Pavlik harness is effective in reducing the knee SA within the treatment group, the Student’s t-test for dependent data will be applied to compare the values of the ultrasound sulcus angle at baseline and two months within each group (treatment and control).

A regression analysis will be performed to determine the relationship between the use of the Pavlik harness and knee SA while accounting for other factors that may affect the knee SA.

The difference in the proportion of participants between the two treatment arms who are classed as a treatment success at two months will be calculated with Fisher’s exact test of independence.

Missing data shall be treated using the multiple imputation procedure to identify patterns in the dataset’s missing data and replace missing values with plausible estimates.

### 2.4. Methods: Monitoring

#### 2.4.1. Harms

To our knowledge, the Pavlik harness has a low-risk profile, although there have been described some rare cases of femoral nerve palsy and avascular necrosis of the femoral head [40].

Regarding security variables, during the follow up, the following will be recorded: confirmation of the good enough position of the Pavlik harness in the revisions carried out every four weeks; note down any treatment difficulties for the patient (such as chafing, maladaptation, crying, etcetera) and identify any concomitant pathologies that have arisen during the period of treatment with the harness to assess whether this involves discomfort or difficulty for the patient and family; interview parents about their ability to adapt in daily activities such as breastfeeding, playing, changing nappies and clothes.

Likewise, a review will be conducted in a face-to-face consultation at 6–9 months in the treatment group to check the correct state of the locomotor apparatus.

#### 2.4.2. Auditing

Frequency and procedures for auditing trial conduct, if any, and whether the process will be independent of investigators and the sponsor will be conducted by the hospital Research Ethics Committee.

## 3. Discussion

TD is a condition identified late and usually requires an invasive treatment when symptomatic.

The possibility of TD being a developmental entity could attain its early diagnosis before patients become symptomatic during adolescence or early adulthood when it is too late for the skeleton to remodel, usually requiring aggressive surgical interventions.

Developmental dysplasia of the hip is a condition that presented a prognosis similar to TD before the introduction of the Pavlik harness. Both conditions have been associated [23,27], so it is plausible to assume they share some common risk factors. In our study, we hypothesize that the breech presentation is a risk factor for TD, as shown in previous human [28] and animal [26] studies. In DDH, a subluxated or dislocated femoral head precipitates the development of a shallow and abnormal acetabulum. Likewise, an extended leg during the breech fetal position could play a similar pathological process as a lack of mechanical forces on the femoral trochlea could precipitate the development of a shallow trochlear sulcus. Thus, we decided to select the newborns in breech presentation as a target group to obtain sonographic data to define radiological values for the dysplastic trochlea. This would serve as a guide for a future screening method that would help identify TD early.

Given that TD may be a developmental process, early diagnosis in the neonatal period could allow for an intervention such as DDH. The Pavlik harness is a treatment option that would maintain the knees bent using the patella to help mold the trochlea in a similar way it is used to make the femoral head mold the acetabulum. Although safe, this treatment is not exempt from certain difficulties in the daily tasks of caring for the newborn, such as nappy changing or dressing. This manifests as a struggle to convince the caregivers of the patient of the future benefits of the Pavlik harness and for compliance, as TD is a condition that usually manifests later in life; thus, the results are not tangible in the short term.

We will conduct a randomized controlled trial to validate the hypothesis that trochlear dysplasia can be treated non-invasively early in life with a Pavlik harness.

Ours is a single-centre study, so the results may not be directly generalized to other centres and the general population. Furthermore, since this is a short-term study, we need to find out the long-term effects of correcting TD with the Pavlik harness or its efficacy compared to other already established treatments for TD, such as MPFL reconstruction or trochleoplasty. We also only assess two months of treatment duration, so there could be more appropriate cost-effective modalities with the Pavlik harness. TD and DDH often present concomitantly, but DDH constitutes an exclusion criterion in our study, so further research will be necessary for these patients to address the best position and regimen of the Pavlik harness for treating both conditions. However, from this study, we can learn about the breech presentation as a risk factor for TD and provide information for future studies on long-term intervention, as ours is the first trial using the Pavlik harness for treating TD.

## Figures and Tables

**Figure 1 jpm-13-00796-f001:**
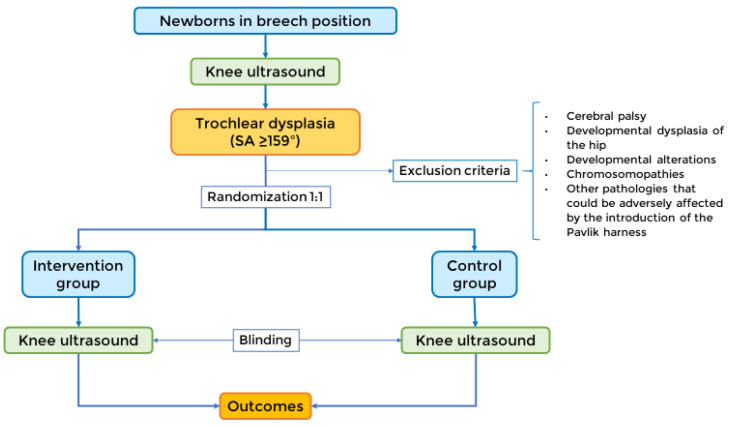
Study design. The newborns in the breech position will be screened with an ultrasound test for dysplastic sulcus angles (≥159°). Then, they will be divided into a 1:1 randomization, either into the intervention group or the control group. A blinded observer will perform a second ultrasound, from which the data for the outcomes will be collected.

**Figure 2 jpm-13-00796-f002:**
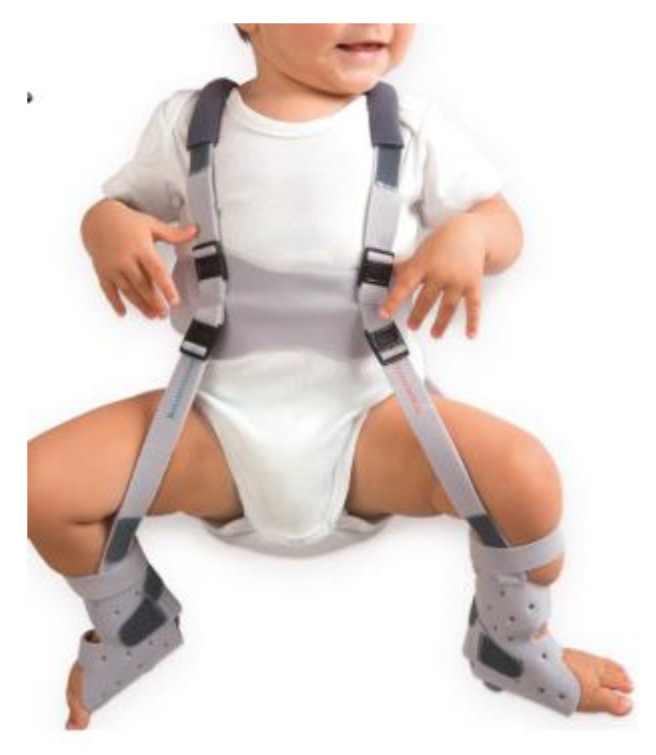
ORLIMAN (Valencia, Spain) Pavlik Harness with the knees placed at a flexion of 90° [35].

**Figure 3 jpm-13-00796-f003:**
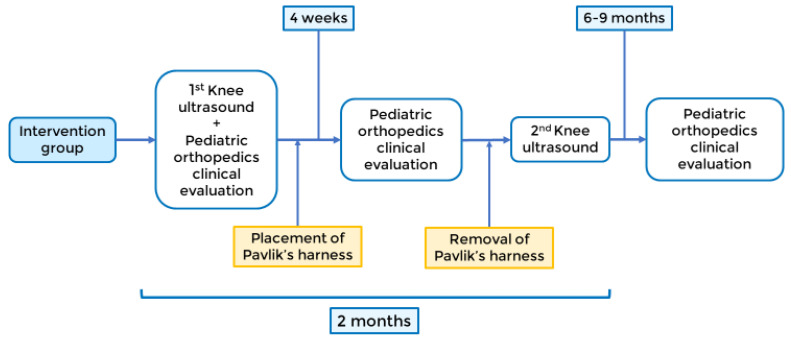
Intervention group timeline.

**Figure 4 jpm-13-00796-f004:**
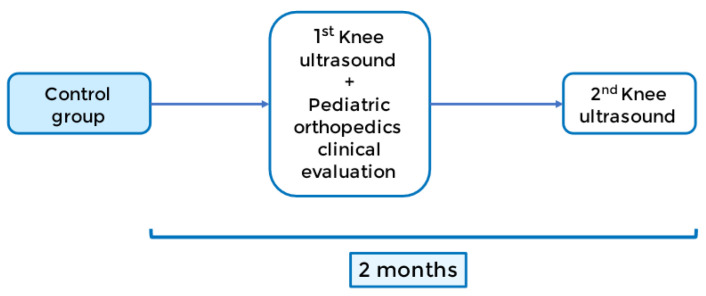
Control group timeline.

## Data Availability

Source documents are all original documents, data, and records. All data collected for this study, both for the preparation of the subject’s clinical history and for the rest of the study documents, will be filed in the participating centres, on paper or in digital format, by the procedures of each centre. A numerical code will identify the data collected for the study, and only the principal investigator/collaborators will be able to relate these data to the patient and their medical history. Access to information on participating subjects will be restricted to the study doctor and authorized collaborating team members. The investigator and the centre will guarantee direct access to the source data or documents to the personnel authorized by the sponsor (monitor, auditor), the health authorities (Spanish Agency of Medicines and Health Products), and the Clinical Research Ethics Committee, when required. Patient data will be collected in a data collection notebook (CRD). The principal investigator or a sub-investigator at the site must ensure the accuracy and completeness of the data recorded and sign the appropriate CRDs. When the database has been deemed complete and accurate, it shall be closed by locking the database. All relevant documents related to the study will be archived according to the requirements of the ICH-GCP, the Commission Directive 2005/28/EC of 8 April 2005, and according to the relevant national laws. The data will be included in a database that must comply with Regulation 679/2016, of 27 April General Data Protection Regulation and Organic Law 3/2018, of 5 December, on personal data protection and guarantee of digital rights. Likewise, the transmission of said data will be carried out with the appropriate security measures in compliance with said regulation. Patients will only be identified by their patient code during documentation and analysis, while the researcher will keep all subject names secret. The investigators are obliged to keep all study data and information confidential and to use these data only in the context of the persons involved in the conduct of the study. Study material or information generated in this study should not be made available to third parties except by official sponsor representatives or regulatory authorities.

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
