# Peer review of "Early Identification and Treatment of Trochlear Knee Dysplasia"

_jpm, 2023, doi:10.3390/jpm13050796_

Round 1

Reviewer 1 Report

The authors presented an interesting study protocol concerning the earl detection and treatment of trochlear knee dysplasia. I would recommend the authors provide more details and clarify statistical tests that will be used to assess/compare outcomes based on the normality of data. Last but not least, the protocol needs to be written in a more scientifically appealing way to readers. 

Author Response

First, we would like to thank you for your comments and for allowing us to address the issues you raise to improve the manuscript’s quality. We have modified those aspects that you have indicated to us.

I will provide a point-by-point response to your comments and objections and indicate the modifications I propose to the manuscript to incorporate your comments.

Review #1 Comments

I would recommend the authors provide more details and clarify statistical tests that will be used to assess/compare outcomes based on the normality of data.

We thank the reviewer for raising this issue. The statistical tests section has been updated in the manuscript providing clarification about the statistical tests used to compare outcomes, as well as analysis of the normality of the data distribution.

Last but not least, the protocol needs to be written in a more scientifically appealing way to readers.

We appreciate the reviewer’s concern. We have added some updates to the manuscript improving wording and clarifying some concepts.

Reviewer 2 Report

This paper suggested the research protocol for infants with trochlear dysplasia.

Interesting topic and protocol is reasonable.

But there was no results. Just protocol.

Authors should investigate the normal value of sulcus angle in the infant.

How did the authors decide the treatment peroid as 2 months?

I think that two months are too short to make the change the trochlear.

probably, the sulcus angle may change with age.

Author Response

First, we would like to thank you for your comments and for allowing us to address the issues you raise to improve the manuscript’s quality. We have modified those aspects that you have indicated to us.

I will provide a point-by-point response to your comments and objections and indicate the modifications I propose to the manuscript to incorporate your comments.

Review #2 Comments

Authors should investigate the normal value of sulcus angle in the infant.

We thank the reviewer for their comments. According to Øye et al., 2015 (1), the mean value of the sulcus angle visualized by ultrasound in newborns has been observed to be 148° ± 5.6°, being an angle of more than 159° defined as dysplastic. However, during this clinical trial, we will also analyze the sulcus angle values to assess the normal values of our particular population.

How did the authors decide the treatment period as 2 months?

We appreciate the reviewer’s interest, and we have updated the manuscript accordingly to clarify this issue. Since this is the first study to evaluate the treatment of trochlear dysplasia with the Pavlik harness, no previous studies assess different treatment modalities. We chose a two-month treatment period based on the most widely adopted treatment regimen for developmental dysplasia of the hip (2).

I think that two months are too short to make the change the trochlear.

We thank the reviewer for raising this issue, and we have added some updates to the manuscript to clarify this concern. Although the timeline has been selected according to the current treatment of developmental dysplasia of the hip, this condition is corrected by remodeling the acetabulum with the help of the mechanical pressure exerted by putting the femoral head in its physiologic position. In the same way, we plan to do with the patella to remodel the trochlea. Both conditions require chondral remodeling to occur, and the cartilaginous tissue in both joints is supposed to follow the same principles of growth, such as Frost’s theory (3). Therefore, although our treatment results for trochlear dysplasia will be analyzed in the clinical trial, according to the current evidence on the treatment duration in developmental dysplasia of the hip, the eight weeks option seems also to be the most cost-effective choice for this investigation.

Probably, the sulcus angle may change with age.

We appreciate the reviewer’s suggestion. Some authors (4–6) have shown that the sulcus angle in their population did not change with age, suggesting that the cartilaginous sulcus angle is a predictor of final trochlea shape.

References

  1. Øye CR, Holen KJ, Foss OA. Mapping of the femoral trochlea in a newborn population: an ultrasonographic study. Acta Radiol [Internet]. 2015 Feb;56(2):234–43. Available from: http://www.ncbi.nlm.nih.gov/pubmed/24553585
  2. Ömeroglu H. Treatment of developmental dysplasia of the hip with the Pavlik harness in children under six months of age: Indications, results and failures. J Child Orthop [Internet]. 2018 Aug 1;12(4):308–16. Available from: http://journals.sagepub.com/doi/10.1302/1863-2548.12.180055
  3. Frost HM. A chondral modeling theory. Calcif Tissue Int [Internet]. 1979 Nov 6;28(3):181–200. Available from: http://link.springer.com/10.1007/BF02441236
  4. Parikh SN, Rajdev N, Sun Q. The Growth of Trochlear Dysplasia During Adolescence. J Pediatr Orthop [Internet]. 2018 Jul;38(6):e318–24. Available from: https://journals.lww.com/01241398-201807000-00008
  5. Trivellas M, Kelley B, West N, Jackson NJ, Beck JJ. Trochlear Morphology Development: Study of Normal Pediatric Knee MRIs. J Pediatr Orthop [Internet]. 2021 Feb 1;41(2):77–82. Available from: http://www.ncbi.nlm.nih.gov/pubmed/33229963
  6. Øye CR, Foss OA, Holen KJ. Minor change in the sulcus angle during the first six years of life: a prospective study of the femoral trochlea development in dysplastic and normal knees. J Child Orthop [Internet]. 2018 Jun 1;12(3):245–50. Available from: http://www.ncbi.nlm.nih.gov/pubmed/29951124